# Molecular evidence supports a genic capture resolution of the lek paradox

Robert J. Dugand[1], Joseph L. Tomkins[1] & W. Jason Kennington[1]

The genic capture hypothesis, where sexually selected traits capture genetic variation in condition and the condition reflects genome-wide mutation load, stands to explain the presence of abundant genetic variation underlying sexually selected traits. Here we test this hypothesis by applying bidirectional selection to male mating success for 14 generations in replicate populations of *Drosophila melanogaster*. We then resequenced the genomes of flies from each population. Consistent with the central predictions of the genic capture hypothesis, we show that genetic variance decreased with success selection and increased with failure selection, providing evidence for purifying sexual selection. This pattern was distributed across the genome and no consistent molecular pathways were associated with divergence, consistent with condition being the target of selection. Together, our results provide molecular evidence suggesting that strong sexual selection erodes genetic variation, and that genome-wide mutation-selection balance contributes to its maintenance.

[1] Centre for Evolutionary Biology, School of Biological Sciences, University of Western Australia, 35 Stirling Highway, Crawley 6009, Australia. Correspondence and requests for materials should be addressed to R.J.D. (email: r.dugand@uq.edu.au)

The maintenance of female preferences for the ornaments of males is a long-standing evolutionary puzzle; why are females persistently choosy when their choice should erode genetic variation among males and preclude any benefits of choice (the lek paradox[1–3])? However, despite being under strong, directional selection[4], the amount of genetic variation underlying sexually selected traits is unexpectedly high[5]. The genic capture hypothesis proposes that high levels of genetic variation can be explained by sexual traits capturing genetic variation in condition[6]. Phenotypic condition is dependent on resource acquisition and allocation[7,8], which, in turn, is thought to be dependent on traits at many loci throughout the genome[9]. Hence, condition could provide a large target for mutations[10] and, as most new mutations are mildly deleterious[11], genetic variation could be maintained in mutation-selection balance across the genome. The persistence of female choice also stands explained, as the offspring of choosy females should inherit a smaller burden of deleterious mutations. Empirical evidence demonstrates that sexually selected traits are condition dependent, that condition dependence has a genetic basis[12–14], and that sexual selection removes deleterious mutations[15–18]. However, molecular evidence for these processes remains largely elusive.

Evolve and resequence[19], where the genomes of experimental populations are sequenced following artificial selection or experimental evolution[19–24], offers a powerful approach for testing evolutionary hypotheses. Applying bidirectional, artificial selection to a focal trait should cause bidirectional changes in allele frequencies at loci under selection. Under the genic capture hypothesis, unconditionally deleterious (and, therefore, rare) alleles that reduce condition are largely expected to contribute to quantitative trait variation. In lines selected for lower fitness, the frequency of rare, deleterious alleles should increase, causing an increase in genetic variance[25], a simple prediction that arises from single-locus models (e.g., see Figure 8.1 in ref.[26]). The opposite should be true for selection for increased fitness. Alternative explanations for the maintenance of genetic variation largely rely on some form of fluctuating or balancing selection (e.g., overdominance[27] or intralocus sexual conflict[28]), where alleles under selection should be at more intermediate frequencies and/or high or low frequencies with variable effects on fitness (see ref.[29]). Although rare alleles may predominantly contribute to trait variation under such alternative models, their effects are expected to be more varied; some rare alleles may increase trait values, whereas other rare alleles decrease trait values. Thus, selection should have more variable effects on genetic variance across loci and the overall genetic variance may even increase in both directions of selection[25].

In a previous experiment, we showed that bidirectional, artificial selection on male mating success generated a strong evolutionary response; lines selected for males that won mate choice trials (success-selected) had higher mating success than lines selected for losers (failure-selected)[18]. The response to selection was correlated with changes in inbreeding depression (in egg-to-adult viability) and sperm competitiveness, potentially indicating that condition, which underlies these traits, may have been the target of sexual selection. Moreover, traits linked to mating success in *D. melanogaster* (e.g., body size[19] and courtship song[24]) are highly polygenic. Thus, our biometric data, together with previous molecular data, align with the genic capture hypothesis.

In this study, we resequence the genomes of our experimentally diverged lines to specifically test the predictions outlined above. We show that success-selected lines have significantly less genetic variation than failure-selected lines, particularly at loci associated with divergence. In addition, we find no evidence that specific sets of genes are associated with divergence in mating success, indicating that condition could be the target of selection. Thus, our

results demonstrate that sexual selection erodes molecular genetic variation and suggest that this genetic variation is maintained in mutation-selection balance across the genome, thereby supporting the genic capture hypothesis.

## Results

**Overview**. To assess the molecular signature of sexual selection in our system, we resequenced the genomes of flies from each of 11 experimental lines (4 success-selected, 4 failure-selected, and 3 control). These lines were generated by applying bidirectional artificial selection to male mating success. Briefly, we selected on mating success by exposing males to mate choice trials, and using males that mounted females to propagate four success-selected lines ($N = 25$ males and 25 females) and males that failed to mount females to propagate four failure-selected lines ($N = 25$ males and 25 females). Four control lines were maintained with flies not exposed to the choice trials (only 3 were used in this study; $N = 25$ males and 25 females). Following 14 generations of selection, we collected 70 females from each line and stored them in ethanol before analysis. We used females so that sample size of the X chromosome was the same as the autosomes. Although we may have missed important variation on the Y chromosome, the primary focus of this study was on the genome-wide effects of selection.

Tissue from the 70 pooled females was homogenised, DNA extracted, and the whole genome sequenced. Pooled sequencing (pool-seq[30]) yields allele counts for the common and rare alleles at each locus, from which we calculated allele frequencies and expected heterozygosity ($H_e$; $2pq$). After quality control and screening out loci where the frequency of rare allele was $< 0.05$ across all 11 lines (within any line, the frequency could be $< 0.05$, but had to be $\geq 0.05$ overall), we identified 55,277 polymorphisms. The allele frequency spectra for all lines can be seen in Supplementary Figure 1a. We then aimed to identify polymorphisms that were associated with divergence (candidate loci) and used these to compare patterns of $H_e$ and Gene Ontology (GO).

**Identifying candidate loci**. We applied two approaches for identifying candidate loci. For our first approach, we calculated the *DiffStat* score at each locus as the minimum allele frequency difference between any combination of success-selected vs. failure-selected lines[19]. The *DiffStat* is zero unless all four lines of one selection regimen have a higher or a lower allele frequency than all four lines from the other selection regimen. We then calculated the maximum allele frequency difference between any combination of lines within a selection regimen (i.e., variation caused by drift). Candidate loci were those where the *DiffStat* was greater than drift. Below, we discuss *Diffstat* loci (i.e., where *Diffstat* > 0), non-*Diffstat* loci (*Diffstat* = 0), and significant *DiffStat* loci (*Diffstat* > drift). For our second approach, we analysed the effect of sexual selection regimen (success-selected vs. failure-selected) on allele frequencies (i.e., the counts of the common and rare allele) at each locus using quasibinomial generalised linear models (GLMs), adjusting for multiple testing using $q$-values with a false discovery rate (FDR) of 0.05[31,32]. We identified 102 significant *DiffStat* loci, 88 significant loci from the GLM approach, and 68 loci significant in both approaches. To be conservative, the following analyses were performed using the 68 significant loci identified from both approaches, which we henceforth refer to as significantly diverged variants (SDVs).

**Genetic variance**. We first evaluated the $H_e$ of success- and failure-selected lines at SDVs. If purifying sexual selection is acting, then the $H_e$ should be lower in success-selected lines for SDVs. Only 3 of the 68 SDVs (4.4%) had higher $H_e$ in success-

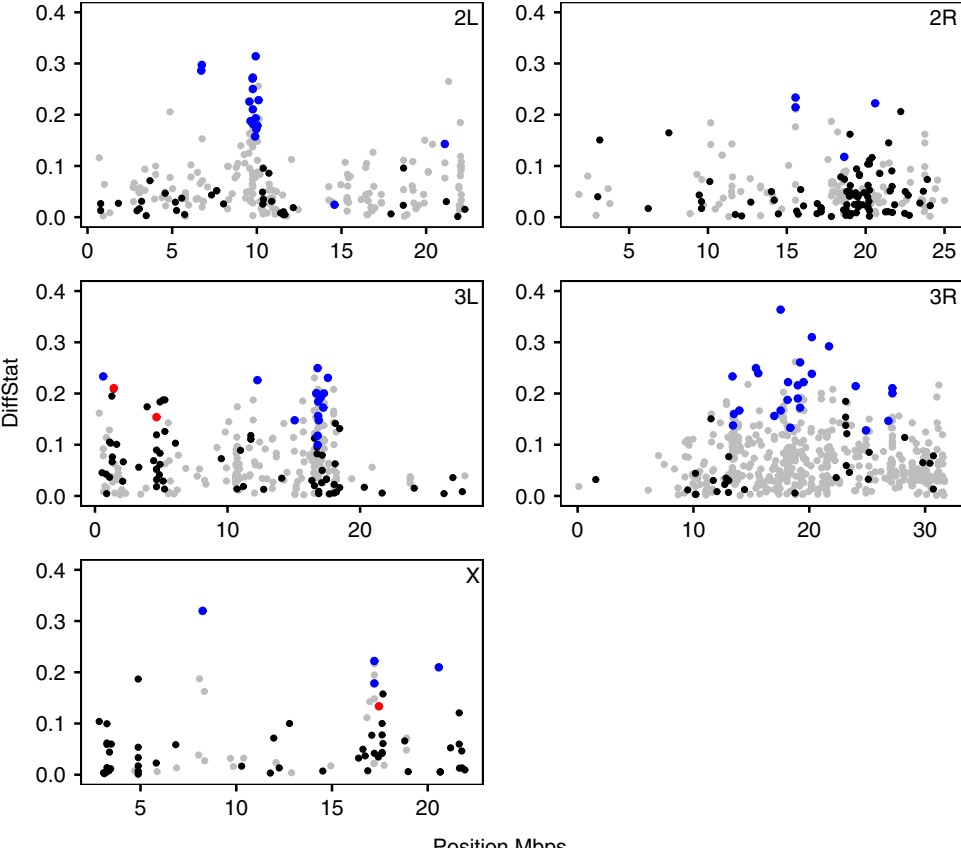

**Fig. 1** *DiffStat* by position across each of the five major chromosome arms. Black points indicate when the expected heterozygosity ($H_e$) of the four success-selected lines is higher than the four failure-selected lines. Red points represent the same pattern for significantly diverged variants (SDVs). Grey and blue points are when the $H_e$ is higher in failure-selected lines for non-SDVs and SDVs, respectively

selected lines. This pattern was consistently strong across all *DiffStat* loci (i.e., including non-SDV loci), where the $H_e$ was higher among success-selected lines for only 272 out of 1363 (20.0%) *DiffStat* loci (Fig. 1; Supplementary Figure 1b). It is noteworthy that variation in the density of SDVs and *DiffStat* loci could be caused by variation in the strength of selection, but could also easily be explained by, e.g., variation in recombination rate along the chromosome arms, which is known to correlate with nucleotide diversity[33]. Below, we refer to the proportion of *DiffStat* loci that have less variation in the success-selected lines (i.e., 0.800 here) as $\mathbf{P_{S<F}}$. These results not only show that success-selected lines have less genetic variation than failure-selected lines, suggesting a role for mutation-selection balance in maintaining genetic variation, but suggest that this effect is widespread, supporting the genic capture hypothesis.

The above analysis does not account for linkage or the density of polymorphisms. Therefore, we compared $H_e$ between selection regimens within 50 kb sliding windows with 10 kb step sizes. We calculated the mean $H_e$ in each sliding window for each experimental line. Our first interest was in identifying sliding windows that had more SDVs than expected by chance (enriched windows), as these represent areas of the genome most likely to be under selection. Of the 12,424 sliding windows across the five chromosome arms, we identified 265 windows with at least 1 SDV and 57 windows that were enriched for SDVs (significant windows). Given the number of sliding windows and the number of significant windows, we then used a resampling approach to test whether enriched windows were evenly distributed across the five chromosome arms. Significant windows were

underrepresented on chromosome 2R, but were otherwise distributed across the other four chromosome arms.

We then compared $H_e$ between selection regimens both genome-wide and within significant windows (see Supplementary Tables 1–2 for $H_e$ statistics for each chromosome arm). Across all sliding windows, $H_e$ was significantly lower in success-selected lines compared with failure-selected lines (Fig. 2a), as determined by a simple *t*-test of the eight line medians ($t = 2.49$; df = 6; $P = 0.047$). This pattern was particularly strong in enriched windows ($t = 9.41$; df = 6; $P = 0.0001$; Fig. 2b). The $H_e$ of control lines was relatively intermediate for all windows and for significant windows (Fig. 2), suggesting that sexual selection increased variation in failure-selected lines and depleted variation in success-selected lines. Together, these results support the main premise of the lek paradox that sexual selection erodes genetic variation and suggest a role for mutation-selection balance in maintaining genetic variation. Moreover, success-selected lines had lower genome-wide (across all sliding windows) gene diversity, consistent with the large mutation target hypothesised under the genic capture hypothesis (evident in Fig. 1), where rare, deleterious mutations from across the genome are implicated in the maintenance of genetic variation[6,34].

**Starting allele frequencies**. Next, we tested whether alleles under selection were likely to have been rare in the base population, as hypothesised if genetic variation is maintained in mutation-selection balance. We used the mean of the three control populations as our estimate of the starting allele frequency at each

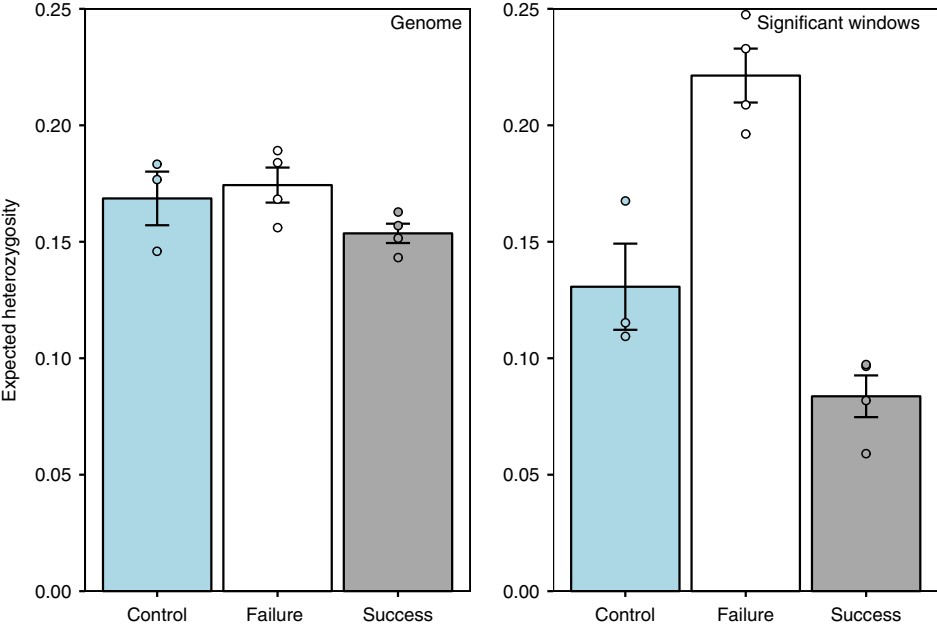

**Fig. 2** Expected heterozygosity ($H_e$) for each selection regimen. $H_e$ was calculated in 50 kb sliding windows with 10 kb step sizes and was calculated across all sliding windows (genome; left) and in windows with two or more significantly diverged variants (significant windows; right). Bars represent the mean ( ± SE) $H_e$ of the line medians (depicted as circles) for each regimen. Control lines are blue, failure-selected lines are white, and success-selected lines are grey. Source data are provided as a Source Data file

locus. We then compared the median starting allele frequency of the SDVs to the genome-wide background. To do so, we randomly resampled (with replacement) 10,000 sets of 68 starting allele frequencies from across the genome and calculated the median allele frequency for each sample, generating a distribution of 10,000 starting allele frequencies. The observed median starting allele frequency of the SDVs was 0.077, significantly lower than the median starting allele frequency, 0.113 (0.085–0.141, 95% confidence intervals) generated by random sampling (i.e., the genome-wide background). This result indicates that SDVs tended to be at low frequency in the starting population, in line with mutation-selection balance.

**Gene Ontology**. We then evaluated whether SDVs were involved with certain molecular pathways. Under the genic capture hypothesis, most of the genome should be involved with the expression of sexually selected traits and, therefore, there is no a priori reason to expect that specific molecular pathways will be involved with divergence in mating success. To assess this, we explored the GO of the SDVs using GOWINDA[35]. We identified 763 GO terms associated with the SDVs; however, none of these GO terms were significant (i.e., no terms were associated with the SDVs more than expected by chance alone). Although this result aligns with the genic capture hypothesis in suggesting that a broad array of molecular pathways (condition) were associated with divergence, it is also possible that the lack of significant GO terms is a simple consequence of having relatively few SDVs to test.

**Genetic drift simulation**. Finally, we performed a series of simulations to test whether our observed results could be reproduced through genetic drift alone. Specifically, we tested whether a comparable reduction in genetic variation could arise if the effective population size ($N_e$) of success-selected lines was smaller than the $N_e$ of the failure-selected lines. Our observed $F_{ST}$ estimate for success-selected lines was slightly higher than for failure-selected lines (0.182 and 0.162, respectively), indicating that $N_e$

may have been smaller in success-selected lines. We simulated neutral evolution for 14 generations with a range of $N_e$ ratios and compared measures of genetic variation ($H_e$ and $P_{S < F}$) and within-regimen genetic divergence ($F_{ST}$). The results from these simulations indicated that differences in $N_e$ between selection regimens could generate similar patterns of $H_e$ and $P_{S < F}$, but not without substantially inflating $F_{ST}$ estimates (see Methods and Supplementary Discussion for a full discussion). Hence, it is highly unlikely that genetic drift alone caused the reduction in $H_e$ in success-selected lines.

## Discussion

Our results provide molecular evidence in support of key evolutionary hypotheses. First, decreased $H_e$ in success-selected lines supports the main premise of the lek paradox that female choice erodes genetic variation. However, we previously demonstrated that deleterious recessive mutations affecting mating success were still segregating in success-selected populations[18], so the biological significance of the extent to which genetic variation has actually been depleted by selection remains unclear. Second, the pattern of $H_e$ that we identified aligns with mutation-selection balance, where the alleles contributing to variation in male mating success in the base populations were rare and consistent in their effects. Our molecular evidence for mutation-selection balance supports our previous finding that success-selected lines were purged of recessive mutations affecting viability, whereas failure-selected lines were not[18]. Third, our results are consistent with the genic capture hypothesis. We found that the median $H_e$ across all sliding windows was lower in the success-selected lines, and that $H_e$ was consistently lower at *DiffStat* loci, at least across the two autosomes. In addition, we found no significant GO terms, demonstrating that no specific molecular pathways have been involved with divergence and suggesting that mutations to any genes can affect mating success through deleterious effects on condition. Together, our results provide comprehensive biometric[18] and molecular evidence in support of a genic capture resolution to the lek paradox.

We found no evidence that intermediate frequency alleles were important to quantitative trait variation underlying mating success, which might have been expected under alternative models for the maintenance of genetic variation. For example, sexual conflict[36–38] is well documented among drosophilids[39–43] and intralocus sexual conflict[44] should maintain alleles at relatively intermediate frequencies through balancing selection. Similarly, molecular evidence has demonstrated that trade-offs between natural and sexual selection can maintain genetic variation through balancing selection[27]. These kinds of antagonistic effects may be prominent in long-standing laboratory populations that are near their adaptive peak[45] and exploring how genetic architecture changes with laboratory adaptation warrants further investigation. Here, using a recently established laboratory population, we found compelling evidence for widespread, negative selection against rare alleles, supporting our previous finding of no trade-offs[18]. More tests such as these are required; we still know very little about the extent to which rare alleles in mutation-selection balance contribute to quantitative trait variation.

The most parsimonious alternative explanation for the pattern of purging that we identified was a smaller $N_e$ of success-selected lines. Indeed, despite equalising the number of breeding individuals across all lines ($N = 50$) following the selection screen, our $F_{ST}$ estimates indicated that the $N_e$ of success-selected lines may still have been slightly lower than the $N_e$ of the failure-selected lines. Although our simulations demonstrate that the smaller $N_e$ could not account for the extent of purging that we identified, it would be worthwhile testing the interacting effects of sexual selection and reduced $N_e$ on molecular (in conjunction with quantitative) genetic variation, as sexual selection does directly reduce $N_e$[46,47]. In particular, evaluating whether or not adaptive variation is eroded by sexual selection in this context would be a valuable avenue to explore. For example, in conservation programs, it might be necessary to weigh potential benefits of sexual selection (purging deleterious mutations[15–18,48]) with the associated cost (loss of genetic variation). In other words, to what extent does sexual selection purge unconditionally deleterious variation and to what extent does it also purge variation with adaptive potential?

Our results highlight the promise of using evolve-and-resequence to test specific hypotheses in sexual selection theory. However, there remain important limitations. First, populations were small and, therefore, the relative effects of genetic drift, as well as the effects of linkage disequilibrium (LD), are enhanced. As a result of both factors, identifying polymorphisms linked to divergence can be compromised. Drift overriding selection will serve to limit our ability to accurately detect SDVs. LD, on the hand, can cause a large number of false positives in evolve and resequence studies[23,49], particularly in small populations when there is strong selection on rare alleles. Our sliding window analysis accounts for short to medium range linkage (10 kb) to some extent, but any effects of long-range LD have not been accounted for and can be substantial[23]. Second, our primary focus was on purifying selection against deleterious (and, therefore, rare) mutations. An initial step in finding loci for analyses was the removal of loci where the frequency of the rare allele was < 0.05, potentially the most important loci. As a result, our evidence for purifying selection is likely to be conservative. Third, males in failure-selected lines had to mate with a female, in order to generate the next generation. Therefore, selection on mating success was weakened, potentially reducing our ability to detect loci strongly associated with variation mating success. Fourth, females used in choice trials were collected from the same experimental line as the competing males. If, e.g., female preferences differ among lines within a regimen, then selection might

be acting on different loci and/or have divergent effects across lines. However, in the final mating success assay, males from each line were competed against standard males for access to standard females (both from the base population). Success-selected males had higher mating success than failure-selected males, so to a large degree the preferences of females in the selection lines aligned with the base population females. In addition, variation in preferences among lines would only serve to increase variance between replicates with regimens. Finally, we used a relatively fresh laboratory stock population in our study. To some extent, experimental lines would still be adapting to the laboratory. Theoretical[50] and empirical evidence[51] suggests that sexual selection can promote adaptation. Therefore, some of the effects that we see may be a consequence of this process. Although this does not undermine our results in any way, it would be interesting to investigate whether the molecular effects of sexual selection differ when populations are near to, or far from, their adaptive peak.

Taken together, our results are molecular evidence for the erosion of genetic variation that sets up the lek paradox and for the genic capture that resolves it. More generally, these results contribute to our understanding of the evolutionary persistence of genetic variation in the face of strong, directional selection.

## Methods

**Overview and selection**. Data manipulation and the majority of statistical analyses were performed in R[52]. Figures were produced using the *ggplot2* package[53].

The stock population of *D. melanogaster* used in this experiment was established by collecting flies en masse from Innisfail, Queensland, in 2012[18]. Stock flies were maintained in cage culture at 25 °C with overlapping generations and a population size of thousands. Artificial selection lines were established approximately ten generations later, after a brief period of laboratory adaptation. We then applied bidirectional selection to male mating success for 14 generations (there were seven generations of selection, three generations of relaxed selection, and a further seven generations of selection) by exposing 7–8-day-old, non-virgin males to binomial mate choice trials and selecting either males that only ever won in mating trials (success-selected; four replicates) or males that only ever lost in mating trials (failure-selected; four replicates). Non-virgin females that were used in mate choice trials were collected from the same line as the males for a given choice trial. Four control lines were maintained with males not exposed to the mate choice trials (only three are represented in this study as one was lost before sequencing). For each line, 25 males (successful/unsuccessful/control) and 25 virgin females were used to establish each generation, making the population size ~50 flies. However, in establishing each generation, we had five replicate vials of five males and five females, which were allowed to mate freely and lay eggs over 24 h. Therefore, within a vial (and indeed between vials), there was the opportunity for selection to act (i.e., selection after the selection assay), both on males and females.

**Sequencing**. After one generation of relaxed selection, we collected 70 females from each selection line and 3 control lines, and stored them in 100% ethanol at −20 °C before DNA extraction. We used females to keep the sample size of the X chromosome and the autosomes the same. Although we may have missed important variation on the Y chromosome, our main test was for mutation-selection balance across the genome. DNA extraction, library construction (Illumina gDNA shotgun library preparation), and genome sequencing were all performed by the Australian Genome Research Facility. Briefly, DNA was extracted from the homogenised tissue of 70 flies (for each line) using NucleoBond Buffer Set IV (Macherey-Nagel). One hundred base pair paired-end sequences were constructed and sequenced using an Illumina HiSeq2500 platform. Image analysis was performed in real time by the HiSeq Control Software v2.2.68 and Real Time Analysis v1.18.66.3. Sequence data were generated using the Illumina bcl2fastq 2.17.1.14 pipeline. Raw fastq sequences were mapped to the *D. melanogaster* reference genome downloaded from FlyBase (flybase.org; r6.08) using BWA version 0.7.12-r1039[54]. Sam files were converted to bam files using samtools[55]. Variant calling was performed using GATK[56], with a UnifiedGenotyper algorithm and ploidy settings equal to 1. The total number of paired-end reads ranged from 21.5 million to 23.2 million across the 11 samples (Supplementary Table 3). Variants different to the reference genome were identified as polymorphisms, which included all insertion/deletion/single nucleotide polymorphisms. Of the 1.4 million polymorphic loci, only ~10% were represented in all 11 populations; all others were discarded. We further excluded reads with mapping qualities < 15 and where the coverage was < 5 or > 250 for any population. For any given population, ~96% of remaining loci had coverage ranging from 15 to 45, with a mean of ~30 (Supplementary Table 3). Finally, we excluded loci where the global frequency of the minor allele was < 5% (i.e., the frequency of the minor allele across all 11

populations), as very rare variants may represent sequencing errors and would not be informative for analyses[24]. This makes the analyses conservative given that the main focus of the study is rare variants. Any other errors are expected to be randomly distributed across the 11 samples. It is noteworthy that each locus did not have to be polymorphic in each population, but there had to be at least five reads from each population for a locus to be included in analyses. After screening, we had 55,277 insertion/deletion/single nucleotide polymorphisms (47,103 single-nucleotide polymorphisms).

**Identifying candidate loci**. We applied two approaches for identifying candidate loci (i.e., variants that may have been under sexual selection). For our first approach, we evaluated whether divergence between selection lines from different regimens exceeded divergence between lines within regimes. We first calculated the frequency of the rare allele at each locus for each population. We then identified loci where the frequency of the minor allele was consistently higher or lower in success-selected lines compared with failure-selected lines. At these loci, we recorded the minimum allele frequency difference between any combination of success-selected vs. failure-selected line, the *DiffStat* for that locus[19]. The *DiffStat* is zero unless all four lines of one selection regimen have a higher or a lower allele frequency than all four lines from the other selection regimen. We identified 1363 *DiffStat* loci (i.e., loci where *DiffStat* > 0). We then calculated the maximum allele frequency difference between any combination of populations within a selection regimen (i.e., variation caused by drift). Candidate loci were those where the *DiffStat* exceeded drift for that locus.

For our second approach, we analysed the effect of selection regimen on the read counts of the common and rare alleles with quasibinomial GLMs, generating *P*-values at each locus with analysis of variances with default *F* tests. We adjusted for multiple testing using *q*-values with a FDR of 0.05[31,32]. Values of $P < 9.50e^{-5}$ were statistically significant. In total, we identified 102 candidates using the first approach and 88 candidates using the second approach (68 were identified by both and these were used in the analyses).

As well as being performed on success vs. failure selection, GLMs were also performed independently for each selection regimen against the control lines to test whether allele frequencies at any loci had significantly diverged from the controls in opposite directions. We identified only one locus (which is not surprising given that population sizes were small and 11 populations were used) where the control lines were significantly divergent from both selection regimens in opposite directions (FBgn0083975; neuroligin 4).

**Hypergeometric test for overrepresentation of SDVs**. We performed a sliding window analysis of $H_e$ ($2pq$; where $p$ and $q$ are the frequencies of the two alleles) to account for differences in the density of loci across the genome. We performed the analysis with 50 kb sliding windows with 10 kb step sizes. We then tested for overrepresentation of SDVs in 50 kb sliding windows[21]. Based on the number of variants and SDVs per chromosome arm, we calculated the probability of identifying *s* SDVs from *n* variants in each window. As with the GLMs, we corrected for genome-wide multiple testing using *q*-values. Given the very low number of SDVs, any windows with two or more SDVs were significantly enriched (i.e., had more SDVs than expected by chance). Overall, we identified 265 windows with at least one SDV and 57 of these had two or more.

We then tested whether or not the 57 significant windows were over- or underrepresented on any of the five major chromosome arms. To do so, we resampled 10,000 sets of 57 windows from across the five major arms and recorded the arm on which the windows were represented. We then compared the location of our 57 significant windows to the null distribution generated from the resampling. Significant windows were underrepresented on chromosome 2R (i.e., fewer windows than the 2.5% confidence interval of six). For the other four chromosome arms, the number of significant windows fell within the 95% confidence intervals.

**Expected heterozygosity ($H_e$) in sliding windows**. We calculated the median $H_e$ across all 50 kb sliding windows for each chromosome arm and then for the whole genome[21] (Supplementary Tables 1–2). We then repeated this for the windows with > 1 SDV (i.e., significant windows).

**Gene Ontology**. We explored the GO of the significant variants to identify the function of gene sets associated with the divergence using GOWINDA[35]. We downloaded GO terms from FuncAssociate3.0 and an annotated genome from FlyBase. We ran the enrichment analysis on the SDVs and identified 763 GO terms. GOWINDA performs permutations tests (100,000 simulations) to calculate the FDR-corrected significance of overrepresentation of GO terms by randomly sampling sets of polymorphisms from the data and recording the genes associated with each polymorphism[35]. The observed GO terms of the SDVs are then compared with this null distribution to identify FDR-corrected significance levels. We identified no significant GO terms.

**Simulating genetic drift**. To test whether our observed patterns for *DiffStats* and $H_e$ could be explained by genetic drift alone, we simulated neutral evolution for 14 generations while manipulating the $N_e$ of the selection lines.

For each simulation, we simulated data for four success-selected lines and four failure-selected lines. For one set of simulations, we held the $N_e$ of the failure-selected lines ($N_{failure}$) constant at 50 and manipulated $N_{success}$ from 20 to 50. For a second set of simulations, we held $N_{failure} = 33$ (~2/3 of the census population size[57]) and $N_{success}$ was 20–33. Our experimental lines comprised 25 males and 25 females; however, we did not control for variation in reproductive success, which may reduce $N_e$. A comparisons of these simulations allowed us to evaluate the effects of different overall $N_e$ on the parameters calculated below.

As loci do not all segregate independently (linked loci will be in LD), we did not simulate data for all 55,277 loci. Instead, we first estimated the extent of LD among linked loci and used this to approximate the number of independently segregating loci in our study. To do so, we assessed the decay (i.e., the increase) of *P*-values surrounding SDVs. We calculated the mean *P*-values of loci within 50 bp non-overlapping sliding windows and concluded that LD had broken down when the curve peaked (Supplementary Figure 3). For all chromosome arms, LD breaks down within 10 kbps. Therefore, the number of loci ($n_{loci}$) that we used in our simulations was determined by effectively sampling one locus every 10 kbps, a total of 12,424 loci. To be conservative, we also ran the simulations on $n_{loci} = 2640$, sampling one locus every 50 kbps.

We did not resequence flies from the base population and, therefore, had no estimate of starting allele frequencies at each locus. As above, we used the mean of the three control lines as our estimate starting allele frequency in the base. For each locus in the simulations, we sampled, with replacement, from these starting allele frequencies. The mean frequency of the rare allele in the control lines was ~0.12.

For each selection line, we then sampled 2N (where N is the population size) alleles with values of 0 or 1 for each locus/line, with the probability of sampling allele 1 equal to the frequency of that allele each generation. To simulate genetic drift, alleles were sampled, with replacement, for 14 generations. In our experiment, there were an additional four generations of genetic drift among our lines when selection was relaxed; however, population sizes during these generations were large (in the order of several hundred) and genetic drift should have been negligible. Thus, we did not include these four generations in our simulations. After generation 14, we sampled k alleles at each locus, where k was identified by sampling (with replacement) a random value from a vector of coverages. Elements of this vector were calculated by taking the average (across all 11 populations) number of reads for each locus from our own data. This final step reflects the random nature of sequencing. We only included loci where the minor allele frequency was ≥ 0.05 and we continued to simulate data until the desired $n_{loci}$ was reached.

Using the final allele frequencies, we then calculated between line $F_{ST}$ values for each regimen (using the *BEDASSLE* R package[58]), the median $H_e$ across all loci for each line, and the *DiffStats* along with their direction (i.e., whether the success- or failure-selected lines had lower $H_e$ at that *DiffStat* locus). We then recorded three statistics: (i) the difference between $F_{ST}$ values between regimens ($DF_{ST}$), (ii) the proportion of diffstat scores where the success-selected lines had less variation ($P_{S < F}$), and (iii) the difference between the mean $H_e$ of the failure-selected lines and the mean $H_e$ of the success-selected lines ($DH_e$). We also recorded the numbers of *DiffStat* and significant *DiffStat* (calculated as above) loci to evaluate how many we expect through drift alone. We repeated this simulation 1000 times for each $n_{loci}$ and for each $N_{success}$, and recorded the mean and 95% confidence intervals for each statistic. All three statistics will be functions of $N_{success}$; we use their relationships to each other to indicate whether genetic drift could cause our results.

## Data availability

The data that support the findings of this study are available in figshare with the identifier doi:10.6084/m9.figshare.7692365. The source data underlying Fig. 2 are provided as a Source Data file.

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

## Acknowledgements
We thank Jacob Berson and the Centre for Evolutionary Biology for constructive discussion. This work was funded by the School of Biological Sciences (The University of Western Australia) and an Australian Research Council Future Fellowship to J.L.T. (FT110100500). R.J.D. was supported by an Australian Postgraduate Award Scholarship through The University of Western Australia.

## Author contributions
R.J.D., W.J.K. and J.L.T. conceived and designed the study. R.J.D. performed the selection. R.J.D. analysed the data and wrote the paper with contributions from all authors.

## Additional information

**Competing interests:** The authors declare no competing interests.

