## [Peer Review File · Nature Communications]

Reviewer #1 (Remarks to the Author):

This is a worthwhile test of the genic capture hypothesis. The study uses the results of replicated selection on male mating success in *Drosophila melanogaster* (success vs failure) combined with genome sequencing to test whether the success lines show the expected pattern of genome-wide reduced genetic variation, as predicted by the hypothesis. The results appear to be robust and to confirm that idea and represent a contribution to knowledge in this area.

The study uses the genome data to test in 2 ways for areas of divergence – first whether the minimum allele frequency difference between selection lines > maximum allele frequency difference between lines within regimes (drift) – and a GLM-type approach to look for candidates. The data from both analyses methods show a signal of increased gene diversity in failure-selected and decreased in success-selected lines – consistent with the genic capture idea.

That said, there are some issues that need to be addressed.

1. It is not clear what would lead the genic capture hypothesis to be rejected. The prediction is reduced genetic variation across the genome as a whole in the success populations. But how would one quantify 'less than genome-wide' coverage? What is the null hypothesis here. Perhaps that can be addressed.
2. There was no signal in the functional enrichment tests and I am not sure whether that was expected or not. How does that feed into the original hypothesis?
3. Unfortunately, because of the compressed format, there isn't much space to give to the design of the selection experiment itself. I'd like to see that remedied by including a new SI section. The design underpins the whole study, so it is important for a reader to be able to understand what has been done. Ione can go to the other work, as did here, to do that, but it doesn't quite feel fair to the reader.
4. As I understand it, the selection is on male mating success and incorporates the choice by non-virgin females between two competing males and then selects the males that can mount females in this scenario. This is selection on males that are attractive in addition to how good those males are at getting matings the ability of those males to mate. Hence, doesn't the selection design incorporate sexual selection in both sexes?
5. Can you add the rationale for the specific design of artificial selection you used? This should include a discussion about the fact that the selection may also include an influence of variation in female choosiness between the regimes. The potential influence of this should be discussed, along with any other factors that varied between the lines (developmental characteristics, time to sexual maturity etc).
6. I think that the test females each generation are from the same lines as the males, hence male mating success is not the same metric across all lines and arises due to an interaction with line females. This may mean that the selection pressure placed on the males from each of the lines may not be the same.
7. All of these points (3-6) about the design of the selection itself could be addressed in a SI section describing the method and rationale in a bit more detail. The important point is to discuss to what extent these factors may have affected the patterns of genetic variation you observed as the test of the overarching hypothesis.

Reviewer #2 (Remarks to the Author):

The study looked for the genomic evidence of genic capture mechanism using replicated experimental evolution fly lines divergently selected for male success and failure. The main expectation was that genome-wide reduction of variation would be observed in success-selected lines due to the more efficient removal of mildly deleterious variants affecting male condition, that were maintained in the base population at the mutation selection-equilibrium. The authors expected that the loss of variation would be particularly pronounced at loci most differentiated between selection regimes. The results were in line with expectations. Unfortunately, in my opinion the results are open to a simpler explanation, which, without

additional evidence, is more likely than the explanation preferred by the authors. This simple explanation is the difference in effective population size (N_e) between the selection regimes. From a short description provided, although the number of founders was the same in each line in each generation, it is likely that higher variation in reproductive success between males, translating into lower N_e , occurred in the success selected lines. If this was the case, I'd expect exactly the result obtained by the authors even without stronger genome-wide selection against deleterious variants in success-selected lines. This does not mean that the scenario proposed by the authors is unlikely, especially given the published phenotypic data and fitness assays, but the evidence at present is not conclusive. At the minimum, to quantify differences in N_e the authors should estimate from allele frequencies the drift distances between lineages. If flies from experimental evolution lines are available, N_e could also be estimated using one or more of numerous available state of the art methods. Alternatively, explicit simulations at various N_e ratios should be performed to investigate the number of significantly differentiated variants in the context of differences in variation between lines from different selection regimes. Such simulations would allow to say whether the obtained results could be a simple consequence of differences in N_e between selection regimes, or whether contribution of selection is necessary to explain the result. An additional, important factor that should be taken into account is the genomic heterogeneity of recombination rate that would cause variation in N_e along the genome. This alone could explain the observed differences in density of differentiated loci in genomic windows. Excellent maps of recombination in the fly genome exist and recombination rate should be taken into account in any such analysis. Before these additional analyses are performed, there is little evidence for the interpretation of the results as the molecular effect of genic capture mechanism. Another important concern I have is the quality of writing, which sometimes is not clear enough, making difficult to understand what exactly was done and how. This may simply be because the text is extremely condensed. It'd benefit from a more standard structure of introduction, results, discussion etc. Methods are very short and I haven't found more extensive description of methods in supplementary materials – they need to be extended so the reader can assess methodology (see Specific comments). The title is not really informative, even if one accepted the interpretation of the results preferred by the authors, because of its ambiguity the title is not effective in communicating the main conclusion of the study.

Specific comments

- l. 15-17 please provide a very brief characteristic of the divergent selection regimes that would be more informative for a general reader
- l. 21 it seems that the authors equate mutation-selection balance with genic capture, which, I believe is incorrect
- l. 51 "or decrease genetic variance" -> "or genetic variance might decrease"
- l. 52 delete "alleles" at the end of the line
- l. 57-59 this is a bit unclear. If I understand correctly, you used all loci with the overall major allele frequency (the average of 11 lines) < 0.95 that were polymorphic in all populations, right? If so I'm not sure that excluding loci that reached fixation in some lines is warranted – at least a rationale for that should be presented.
- l. 66 the rationale that the meaningful candidates can be identified by using DiffStat as described here is not convincing – one can expect many loci showing the minimum allele frequency differences between selection regimes exceeding the maximum difference between lines within regime due to drift alone (the randomization test described in l. 209 – 211 is a step in the right direction, but explicit simulations would be much better)
- l. 75-86 as explained above, without information on effective population sizes and explicit simulations modeling the situation under corresponding drift strength, these results in my opinion do not provide evidence for the removal of deleterious variation by sexual selection. Lower variation can simply result from stronger drift in success-selected lines.
- l. 87-97 also here the most parsimonious explanation is heterogeneous neutral divergence due to differences in N_e across genome because of variation in recombination rate. In this analysis local recombination rate should be taken into account as covariate.
- l. 131-142 I don't understand the link between the results and conclusions here
- l. 188 and ff - please provide information what was the coverage per pool
- l. 190 please add information about methods used for detecting polymorphism in poolseq data
- l. 207 please explain how exactly were control populations treated
- l. 232 and ff – please explain that gene diversity is the same as expected heterozygosity, which is

a more commonly used term

Reviewer #3 (Remarks to the Author):

The manuscript titled "Molecular evidence for genome-wide mutation-selection balance and the paradox of the lek" evaluated the genic capture hypothesis in lines of *Drosophila* that have been selected for either male mating success or failure. I found the central question proposed by the authors to be valuable given that understanding how trait variation is maintained in the face of strong selection is an important question in evolutionary biology. The authors also used an interesting approach to address this question. However, I found their methods somewhat hard to follow and I have a few questions regarding statistical approaches that were used. One of my biggest questions is why were only females used in the study if the authors were specifically focusing on lines selected for differences in male mating success. I would assume that there is some sexual conflict at certain regions of the genome that may be confounding some of these findings. My specific comments are below:

Lines 19-20: "... and fall into a range of molecular pathways"

But the authors do not fully discuss these molecular pathways since no significant GO terms were enriched by their SDVs.

Line 52: "... If variation is caused by alleles at intermediate-frequency alleles"

Remove the second instance of 'alleles'.

Line 75: I understand that the patterns obtained were consistent between both the DiffStat and GLM approaches, but why not focus on the 42 regions that were represented in both methods to further explore how they differ between the success-selected and failure-selected populations. This seems to be a more conservative approach given that these regions were shown to differ using two different methods.

Lines 76-80: I'm slightly confused as to how the DiffStat scores were used. The authors initially discuss loci where the maximum allele frequency difference between lines within a regimen was lower than the minimum allele frequency difference between selection lines from different regimens and this resulted in 57 candidate polymorphisms. However, they then mostly discuss the regions where the DiffStat scores was > 0 (in other words, where the Diffstat score for all four of the selected lines was a higher or lower allele frequency than all four of the other selected line). This resulted in 1,366 scores being less than 0. Are the 57 candidate loci also included in the regions with DiffStat scores >0 ? I'm assuming they are but I don't understand why the remainder of this section focuses on the 1,366 scores rather than the 57 candidate loci identified.

Line 123-124: How does the median starting allele frequency for SDVs in the control population compare to the allele frequency after selection in the success-selected and failure-selected lines?

Lines 137-142: But if selection is strong enough to where you see erosion of genetic diversity with successful-selected lines, shouldn't this be strong enough to pick up gene that may be influencing these behaviors? Is it possible by only focusing on females, you may be missing this information if there is significant sexual conflict at sights that may increase male mating success?

Lines 177: What was the justification in using only females for this work? Given that the study specifically focuses on selection driven by male mating success it seems odd to me that only females were used for sequencing.

Line 218-219: "We identified only one locus where the control populations were significantly divergent from both selection regimes I opposite directions."

For the 88 and 57 loci identified using GLM and DiffStat, does this mean they only differed between the two selection lines or did they also differ between the selection lines and control populations? This needs to be clear in the manuscript as it give some indication as to how strong of selection pressure may be acting on these lines.

Line 219: 3R-20224662, FBgn0083975?

It would be helpful to the reader to state that this gene is neuroligin 4.

Extended Data Figure 1b: I'm really confused about the interpretation of figure because shouldn't successful selected lines purge these low frequency alleles if they a negative impact on mating success, thus decreasing genetic variation?

Reviewer #1 (Remarks to the Author):

This is a worthwhile test of the genic capture hypothesis. The study uses the results of replicated selection on male mating success in *Drosophila melanogaster* (success vs failure) combined with genome sequencing to test whether the success lines show the expected pattern of genome-wide reduced genetic variation, as predicted by the hypothesis. The results appear to be robust and to confirm that idea and represent a contribution to knowledge in this area.

The study uses the genome data to test in 2 ways for areas of divergence – first whether the minimum allele frequency difference between selection lines > maximum allele frequency difference between lines within regimes (drift) – and a GLM-type approach to look for candidates. The data from both analyses methods show a signal of increased gene diversity in failure-selected and decreased in success-selected lines – consistent with the genic capture idea.

That said, there are some issues that need to be addressed.

1. It is not clear what would lead the genic capture hypothesis to be rejected. The prediction is reduced genetic variation across the genome as a whole in the success populations. But how would one quantify 'less than genome-wide' coverage? What is the null hypothesis here. Perhaps that can be addressed.

The reviewer makes a good point that we did not adequately set up the predictions for genic capture or discuss the results in enough detail to show how our results support the hypothesis. We have amended this in the revised version of the manuscript. We have set out predictions in paragraphs 3 and 4 of the introduction, and have expanded our explanation for the support of the genic capture hypothesis in the first paragraph of the discussion (as well as throughout the results). Our main points are that the patterns that we see are largely consistent across the chromosome arms, that genetic variance is depleted when comparing the gene diversity at the genome level, and that we found no evidence for significant genes sets associated with divergence. While it is hard to say what 'less than genome-wide' might be, these results all conform to the genic capture hypothesis.

2. There was no signal in the functional enrichment tests and I am not sure whether that was expected or not. How does that feed into the original hypothesis?

Under the genic capture hypothesis, mutations to any gene should reduce mating success by decreasing condition. The hypothesis, therefore, is that we didn't expect any significant GO terms. In contrast, if variation in male mating success was determined by a few key traits, then we might expect significant GO terms. This is now explained in the first paragraph of the discussion.

3. Unfortunately, because of the compressed format, there isn't much space to give to the design of the selection experiment itself. I'd like to see that remedied by including a new SI section. The design underpins the whole study, so it is important for a reader to be able to understand what has been done. One can go to the other work, as did here, to do that, but it doesn't quite feel fair to the reader.

We agree that there was not enough detail about the selection experiment itself. We have expanded our methods section to include details about the experimental design and discuss important points about the experimental design. See SM1 and the second last paragraph of the discussion.

4. As I understand it, the selection is on male mating success and incorporates the choice by non-virgin females between two competing males and then selects the males that can mount females in this scenario. This is selection on males that are attractive in addition to how good those males are at getting matings the ability of those males to mate. Hence, doesn't the selection design incorporate sexual selection in both sexes?

5. Can you add the rationale for the specific design of artificial selection you used? This should include a discussion about the fact that the selection may also include an influence of variation in female choosiness between the regimes. The potential influence of this should be discussed, along with any other factors that varied between the lines (developmental characteristics, time to sexual maturity etc).

Selection could have been on male-male competition, attractiveness etc. We chose to select on mating success so that it incorporated all components of precopulatory sexual selection in a relatively realistic scenario (discussed in detail in Dugand et al 2018). The downside is that we do not know how selection is really working, but in the context of the previous experiment, that wasn't important. Again, here, how selection is acting does not undermine our evidence that sexual selection purges molecular genetic variation. Additionally, when we measured male mating success following artificial selection, males from our experimental lines were competed against stock population males for access to stock population females. Thus, our measures of divergence are all about divergence in

male mating success, not about differences in female choosiness or otherwise. We have expanded our methods section, and included discussion about potential key 'problems' with our experiment (discussion second last paragraph).

6. I think that the test females each generation are from the same lines as the males, hence male mating success is not the same metric across all lines and arises due to an interaction with line females. This may mean that the selection pressure placed on the males from each of the lines may not be the same.

We have added a section in the discussion (second last paragraph) to address this point.

"Fourth, females used in choice trials were collected from the same experimental line as the competing males. If, for example, female preferences differ among lines within a regimen, then selection might be acting on different loci and/or have divergent effects across lines. However, in the final mating success assay, males from each line were competed against standard males for access to standard females (both from the base population). Success-selected males had higher mating success than failure-selected males, so to a large degree the preferences of females in the selection lines aligned with the base population females. Additionally, variation in preferences among lines would only serve to increase variance between replicates with regimens."

7. All of these points (3-6) about the design of the selection itself could be addressed in a SI section describing the method and rationale in a bit more detail. The important point is to discuss to what extent these factors may have affected the patterns of genetic variation you observed as the test of the overarching hypothesis.

We hope that we have now provided sufficient detail about the experimental design and how it might influence our results. The most important point, raised by reviewer 2, relates to regimen-specific effects on effective population size (see our comments below).

Reviewer #2 (Remarks to the Author):

The study looked for the genomic evidence of genic capture mechanism using replicated experimental evolution fly lines divergently selected for male success and failure. The main expectation was that genome-wide reduction of variation would be observed in success-selected lines due to the more efficient removal of mildly deleterious variants affecting male condition, that were maintained in the base population at the mutation selection-equilibrium. The authors expected that the loss of variation would be particularly pronounced at loci most differentiated between selection regimes. The results were in line with expectations.

Unfortunately, in my opinion the results are open to a simpler explanation, which, without additional evidence, is more likely than the explanation preferred by the authors. This simple explanation is the difference in effective population size (N_e) between the selection regimes. From a short description provided, although the number of founders was the same in each line in each generation, it is likely that higher variation in reproductive success between males, translating into lower N_e , occurred in the success selected lines.

In many experimental evolution studies, the N_e of sexual selection treatments will be lower. However, in our study, 25 males that passed through selection screens were then placed with virgin females. The opportunity for sexual selection after the selection screen was equivalent between success-selected and failure-selected lines, and therefore there is no reason to expect that the N_e of the success-selected populations would be lower than the failure-selected populations with our design. This was not something that we discussed very clearly in the previous ms, and therefore we hope that we have clarified things now. Even so, the possibility of differences in N_e is now a substantial part of the ms, and we find no evidence to support drift as a factor driving our results (see below).

If this was the case, I'd expect exactly the result obtained by the authors even without stronger genome-wide selection against deleterious variants in success-selected lines. This does not mean that the scenario proposed by the authors is unlikely, especially given the published phenotypic data and fitness assays, but the evidence at present is not conclusive. At the minimum, to quantify differences in N_e the authors should estimate from allele frequencies the drift distances between lineages. If flies from experimental evolution lines are available, N_e could also be estimated using one or more of numerous available state of the art methods. Alternatively, explicit simulations at various N_e ratios should be performed to investigate the number of significantly differentiated variants in the context of differences in variation between lines from different selection regimes. Such simulations

would allow to say whether the obtained results could be a simple consequence of differences in N_e between selection regimes, or whether contribution of selection is necessary to explain the result. This is an excellent point and we are very pleased to be able to incorporate these simulations into our study. We have simulated neutral evolution in replicate populations and manipulated the N_e of success-selected populations as suggested. We are now able to show that our main result – that sexual selection applies negative selection against rare alleles across the genome – is robust to differences in population sizes between selection regimes. The extent of the reduction in genetic variance in success-selected lines is substantially greater than expected by any drift scenario, demonstrating that sexual selection erodes genetic variation. Including these simulations greatly improves the quality of our manuscript.

An additional, important factor that should be taken into account is the genomic heterogeneity of recombination rate that would cause variation in N_e along the genome. This alone could explain the observed differences in density of differentiated loci in genomic windows. Excellent maps of recombination in the fly genome exist and recombination rate should be taken into account in any such analysis. Before these additional analyses are performed, there is little evidence for the interpretation of the results as the molecular effect of genic capture mechanism.

We have commented on the possibility that variation in the density of SDVs could be caused by variation in recombination rate (paragraph 4 in the results), and, therefore, removed our statement that higher densities of SDVs means stronger selection. Given the broad, consistent patterns that our data show, it is hard to see how variation in recombination rate would strongly influence our results, particularly in any regimen-specific way.

Another important concern I have is the quality of writing, which sometimes is not clear enough, making difficult to understand what exactly was done and how. This may simply be because the text is extremely condensed. It'd benefit from a more standard structure of introduction, results, discussion etc. Methods are very short and I haven't found more extensive description of methods in supplementary materials – they need to be extended so the reader can assess methodology (see Specific comments).

We have substantially revised the manuscript to improve the quality of writing. Our revised manuscript is in the standard format and contains more comprehensive explanations of important points.

The title is not really informative, even if one accepted the interpretation of the results preferred by the authors, because of its ambiguity the title is not effective in communicating the main conclusion of the study.

We have changed our title: "Molecular evidence for a genic capture resolution of the lek paradox"

Specific comments

I. 15-17 please provide a very brief characteristic of the divergent selection regimes that would be more informative for a general reader

We have provided information of this in the abstract as advised. "Bidirectional selection was applied to male mating success for 14 generations in replicate populations."

I. 21 it seems that the authors equate mutation-selection balance with genic capture, which, I believe is incorrect

We did not intend to and hope that this is now amended in the revised ms.

I. 51 "or decrease genetic variance" -> "or genetic variance might decrease"

I. 52 delete "alleles" at the end of the line

This section was deleted in favour of a more concise explanation, so these errors are no longer present.

I. 57-59 this is a bit unclear. If I understand correctly, you used all loci with the overall major allele frequency (the average of 11 lines) < 0.95 that were polymorphic in all populations, right? If so I'm not sure that excluding loci that reached fixation in some lines is warranted – at least a rationale for that should be presented.

Loci that reached fixation in some lines were included. The confusion was due to our poor word choice. This should now be clear (second paragraph of the results).

I. 66 the rationale that the meaningful candidates can be identified by using DiffStat as described here

is not convincing – one can expect many loci showing the minimum allele frequency differences between selection regimes exceeding the maximum difference between lines within regime due to drift alone (the randomization test described in l. 209 – 211 is a step in the right direction, but explicit simulations would be much better)

As described above, we performed simulations to mimic neutral evolution. From these simulations, we also generated confidence intervals for the number of significant DiffStat loci that we would expect to find by chance. Our observed value falls outside of the 95% intervals expected by chance (see paragraph 3 of the SM3). This was a valuable inclusion in the ms.

l. 75-86 as explained above, without information on effective population sizes and explicit simulations modeling the situation under corresponding drift strength, these results in my opinion do not provide evidence for the removal of deleterious variation by sexual selection. Lower variation can simply result from stronger drift in success-selected lines.

l. 87-97 also here the most parsimonious explanation is heterogeneous neutral divergence due to differences in N_e across genome because of variation in recombination rate. In this analysis local recombination rate should be taken into account as covariate.

See previous comments for details of simulations.

l. 131-142 I don't understand the link between the results and conclusions here

We have expanded our discussion which includes an explanation of why no significant GO terms might conform to the genic capture hypothesis (paragraph 1 of discussion). This should also be clearer in the results section (second last paragraph of the results).

l. 188 and ff - please provide information what was the coverage per pool

We have included a table of coverage (Supplementary Table 3). ~96% of loci had coverage ranging from 15-45.

l. 190 please add information about methods used for detecting polymorphism in poolseq data

We have included a sentence that explains that polymorphisms are detected by identifying variants that are different to the reference genome, that have at least two alleles, and where the frequency of common allele is <0.95 across all populations.

l. 207 please explain how exactly were control populations treated

We have expanded the methods section. This should now be clear.

l. 232 and ff – please explain that gene diversity is the same as expected heterozygosity, which is a more commonly used term

We have favoured expected heterozygosity (H_e) in our revised ms.

Reviewer #3 (Remarks to the Author):

The manuscript titled “Molecular evidence for genome-wide mutation-selection balance and the paradox of the lek” evaluated the genic capture hypothesis in lines of *Drosophila* that have been selected for either male mating success or failure. I found the central question proposed by the authors to be valuable given that understanding how trait variation is maintained in the face of strong selection is an important question in evolutionary biology. The authors also used an interesting approach to address this question. However, I found their methods somewhat hard to follow and I have a few questions regarding statistical approaches that were used.

One of my biggest questions is why were only females used in the study if the authors were specifically focusing on lines selected for differences in male mating success. I would assume that there is some sexual conflict at certain regions of the genome that may be confounding some of these findings.

Our specific focus was on genome-wide mutation-selection balance. It would be interesting to assess the Y chromosome, but as this was not a main focus, and given that we wanted to keep the sample size of the X chromosome and the autosomes equivalent, we chose to use females.

My specific comments are below:

Lines 19-20: “... and fall into a range of molecular pathways”

But the authors do not fully discuss these molecular pathways since no significant GO terms were

enriched by their SDVs.

Poor word choice on our behalf, we have changed this in the abstract and results.

Line 52: "... If variation is caused by alleles at intermediate-frequency alleles"

Remove the second instance of 'alleles'.

This section was modified.

Line 75: I understand that the patterns obtained were consistent between both the DiffStat and GLM approaches, but why not focus on the 42 regions that were represented in both methods to further explore how they differ between the success-selected and failure-selected populations. This seems to be a more conservative approach given that these regions were shown to differ using two different methods.

We have re-analysed all of our data, focusing on 68 variants that were identified by GLMs and by the significant DiffStat approach. To make our approaches (GLMs and by the significant DiffStat) comparable, we have excluded the control populations from the DiffStat analysis. As such, there were 102 significant DiffStat loci (previously 57) and 68 loci significant in both approaches (previously 42).

Lines 76-80: I'm slightly confused as to how the DiffStat scores were used. The authors initially discuss loci where the maximum allele frequency difference between lines within a regimen was lower than the minimum allele frequency difference between selection lines from different regimens and this resulted in 57 candidate polymorphisms. However, they then mostly discuss the regions where the DiffStat scores was > 0 (in other words, where the Diffstat score for all four of the selected lines was a higher or lower allele frequency than all four of the other selected line). This resulted in 1,366 scores being less than 0. Are the 57 candidate loci also included in the regions with DiffStat scores > 0 ? I'm assuming they are but I don't understand why the remainder of this section focuses on the 1,366 scores rather than the 57 candidate loci identified.

We have expanded this section for clarity. We discuss the 1363 DiffStat loci because, even though lots of them will be caused by drift, they are the most likely to have been involved with divergence. Even though they are not all significant (only 102 were), they still show the same pattern as the significant loci.

Line 123-124: How does the median starting allele frequency for SDVs in the control population compare to the allele frequency after selection in the success-selected and failure-selected lines?

This information can be gleaned from Figure 1B (expected heterozygosity is simply related to allele frequency when there are only two alleles at a loci) as well as Supplementary Figure 1 (where most alleles are at higher frequencies in the failure-selected lines, and purged in success-selected lines). The expected heterozygosity goes up a lot in the failure-selected lines, and down a bit in the success-selected lines.

Lines 137-142: But if selection is strong enough to where you see erosion of genetic diversity with successful-selected lines, shouldn't this be strong enough to pick up gene that may be influencing these behaviors? Is it possible by only focusing on females, you may be missing this information if there is significant sexual conflict at sights that may increase male mating success?

It is a good point that, given that the lines responded, that we might have picked up genes associated with the response. We suggest that the lack of significant GO terms indicates that a broad array of gene functions were associated with divergence, which aligns with condition theory. We do not agree that we "only focus on females" since males also have autosomes, but agree that we may miss important information on the Y chromosomes, and we mention this in the ms (see methods).

Lines 177: What was the justification in using only females for this work? Given that the study specifically focuses on selection driven by male mating success it seems odd to me that only females were used for sequencing.

Our study actually aimed to test for a genome-wide spread of mutations (which we justify because we previously show that mutations were purged in success-selected lines); we sequenced females so that the sample size of the X chromosome was the same as the autosomes. While we did select specifically on males, the effect of selection on the genome is the most important point, because that will determine the fitness of both males and females.

Line 218-219: "We identified only one locus where the control populations were significantly divergent from both selection regimes I opposite directions."

For the 88 and 57 loci identified using GLM and DiffStat, does this mean they only differed between the two selection lines or did they also differ between the selection lines and control populations? This needs to be clear in the manuscript as it give some indication as to how strong of selection pressure may be acting on these lines.

There are 8 selection lines, four in each selection regimen. The controls were not included in the success v failure analyses, but GLMs were performed for controls versus success and controls versus failure. This section is in SM3, and should now be clearer.

Line 219: 3R-20224662, FBgn0083975?

It would be helpful to the reader to state that this gene is neurologin 4.

Added as suggested.

Extended Data Figure 1b: I'm really confused about the interpretation of figure because shouldn't successful selected lines purge these low frequency alleles if they a negative impact on mating success, thus decreasing genetic variation?

This figure has been modified so that there is a 0 freq pool, previously that was included in the <0.1 pool. This should remove some confusion.

Reviewer #1 (Remarks to the Author):

I am happy with this revised version and thank the authors for engaging so positively and constructively with the review process. I am satisfied with the revisions that have been made in response to my original comments and to those of the other two reviewers.

Reviewer #2 (Remarks to the Author):

The revised version is much improved – writing is clearer and easier to follow. Extensive simulations that model a possible effect of differences in N_e between selection regimes on the results have been added. While I do appreciate these efforts and think that the evidence for the effect of purifying selection on deleterious mutations in success-selected lines is stronger now, more realistic simulations could be performed. While reasonable ratios of success/failure N_e have been simulated, the reference N_e of 50 individuals is clearly an overestimate. In this case probably not only the ratio but also the absolute value of the reference N_e matter. Wouldn't it really be possible to estimate N_e of each line using molecular markers, for example from the loss of heterozygosity in the course of the experiment? That would be information crucial for designing realistic simulations. If estimation is not possible, informed guess could be based on results of earlier studies. For example in the classical Buri (1956) experiment where the ratio of the effective to census population size in small lab populations of *Drosophila* was estimated as ca 2/3 (i.e. ca. 33 individuals in the present case). Another quite important point that only D_{Fst} values from simulations are reported, while one would like to know whether F_{st} values from simulations were comparable to the estimates from empirical data. Finally, more information about detection of polymorphism and estimation of allele frequencies is needed (l. 322 and ff.). What was the approach and software used?

Minor points

While it's clear from MM that only variants that were polymorphic in all lines were included, it should be better explained elsewhere in text (l. 91-94)

l. 61-62 here one more sentence of explanation would help the reader to make a causal link between these two sentences

l 86 remove "that"

l. 98 do I understand correctly that you use "loci" because these could be SNPs or small indels – how many of each of these were there?

l. 125 here again a bit of explanation would help the reader the reader to understand the logic

l. 135 you mean evenly distributed?

l. 223-226 I don't understand this reasoning that variation in mating success between males from failure selected lines would be smaller than between males from success selected lines

l. 229-31 the argument about F_{st} may indeed be convincing but please note that in success-selected lines selection is expected to act in the same direction on deleterious variants, which would prevent

differentiation via drift in these variants, thus slowing down the increase of overall F_{st} (this effect may be small as these are mostly low frequency polymorphisms, but still). In the failure-selected lines on the contrary, frequencies of both alleles in these loci are more free to vary via drift.

I. 237 the claim that selection depletes variation without a reduced N_e is too bold in my opinion

I. 247-8 wouldn't that be against the predictions of genic capture hypothesis?

I. 249-51 these are rather weakly supported speculations

compare I. 84 and 308: four vs one generation of relaxed selection. Does this mean that selection/relaxed selection episodes were intermixed during 14 generations of selection? Please provide more explanation.

I. 315 -316 libraries were constructed and sequenced – what was library the type?

I. 325 loci -> reads

I. 367-74 this reasoning may have allowed you to estimate the ration, but it'd be much cleaner if you had estimates of actual N_e in your lines

I. 430 I don't understand why you think this was a conservative approach – you probably simulated weaker overall drift than in the experiment so I wouldn't call this approach conservative.

Reviewer #3 (Remarks to the Author):

I would like to thank the authors for the revised edition of the manuscript. The majority of my initial comments have been addressed, but I have a few additional suggestions/questions.

Line 124: 'density of SDVs and DiffStat loci'. I'm assuming this is referring to the density along the chromosome arms from Figure 1? If so, please state this or move this statement so that it immediately follows the sentence where you refer to Figure 1.

Line 183: Since no other section of the supplemental methods is identified I suggest removing the (SM7) from the sentence.

While I understand that no GO terms showed significant enrichment, it would be nice to have an idea what the 763 GO terms were associated with. Is it possible to put this information in a supplemental table? Also, in my experience 57 loci is a relatively small data set for enrichment analysis and I'm wondering if this is why you didn't observe enriched pathways.

It seems as though the majority of the discussion focuses on the new analyses associated with examining how variation in N_e might be impacting the results. However, all of these results have been placed in a supplemental file. If this is now the main point of the manuscript, these data need to be moved to the main body of the paper.

I also find that the discussion is rather narrow in its focus, particularly regarding how N_e might impact the results. I would suggest the authors try to broaden this section (and perhaps the introduction) and discuss how these data support the genetic capture hypothesis in contrast to alternative hypotheses regarding the maintenance of genetic variation under sexual selection.

Line 318: should be '100 bp paired-end sequences were constructed and sequenced using an Illumina HiSeq2500 platform'.

Line 325: What version of bwa?

Lines 321: "total number of paired-end reads ranged from 21.5 – 23.2 million." From the sup table it seems as though you are averaging around 43 million per line. If the reads in the table are the total per sample (including forward & reverse), please state that to prevent confusion.

Lines 369: "The expected number of DiffState loci....."

I think that this is an important finding in regard to how the data set are likely associated with selection rather than just drift and should be included in the main results section.

Reviewers' comments:

Reviewer #1 (Remarks to the Author):

I am happy with this revised version and thank the authors for engaging so positively and constructively with the review process. I am satisfied with the revisions that have been made in response to my original comments and to those of the other two reviewers.

Reviewer #2 (Remarks to the Author):

The revised version is much improved – writing is clearer and easier to follow. Extensive simulations that model a possible effect of differences in N_e between selection regimes on the results have been added. While I do appreciate these efforts and think that the evidence for the effect of purifying selection on deleterious mutations in success-selected lines is stronger now, more realistic simulations could be performed. While reasonable ratios of success/failure N_e have been simulated, the reference N_e of 50 individuals is clearly an overestimate. In this case probably not only the ratio but also the absolute value of the reference N_e matter. Wouldn't it really be possible to estimate N_e of each line using molecular markers, for example from the loss of heterozygosity in the course of the experiment? That would be information crucial for designing realistic simulations. If estimation is not possible, informed guess could be based on results of earlier studies. For example in the classical Buri (1956) experiment where the ratio of the effective to census population size in small lab populations of *Drosophila* was estimated as ca 2/3 (i.e. ca. 33 individuals in the present case).

Another quite important point that only D_{FST} values from simulations are reported, while one would like to know whether F_{ST} values from simulations were comparable to the estimates from empirical data.

The F_{ST} values from our original simulations (where $N_{failure}=50$) were in the range of ~0.16, compared to 0.162 for failure-selected lines and 0.182 for success-selected lines. Hence, the N_e of our lines does appear to be slightly lower than 50. Therefore, we repeated the simulations with $N_e=33$, as suggested. The F_{ST} values at $N_e=33$ were ~0.22, suggesting that the N_e values of our lines were in the range of 33-50. We have included both sets of simulations in our revised manuscript. The outcomes are much the same, and support our previous conclusion that drift has not caused our results.

Finally, more information about detection of polymorphism and estimation of allele frequencies is needed (l. 322 and ff.). What was the approach and software used?

We have added information about the software used and the steps taken in detecting polymorphism. (l299-305)

Minor points

While it's clear from MM that only variants that were polymorphic in all lines were included, it should be better explained elsewhere in text (l. 91-94)

We did not only include variants that were polymorphic in all lines. We included variants that were polymorphic across all lines.

l. 61-62 here one more sentence of explanation would help the reader to make a causal link between these two sentences

We have amended this section for clarity (l57-65).

l 86 remove "that"

Fixed.

l. 98 do I understand correctly that you use "loci" because these could be SNPs or small indels – how many of each of these were there?

Yes, there were SNPs and indels. This information has been included in the methods: "After screening, we had 55,277 insertion/deletion/single nucleotide polymorphisms (47,103 SNPs)" (l317-318).

l. 125 here again a bit of explanation would help the reader the reader to understand the logic

We have moved this sentence to above (as suggested by reviewer 3) so that it now flows logically from the previous sentence (l121-124).

l. 135 you mean evenly distributed?

Fixed.

l. 223-226 I don't understand this reasoning that variation in mating success between males from failure selected lines would be smaller than between males from success selected lines

The sentence actually suggested the opposite: "that variation would be greater among failure-selected males owing to their relatively poor mating success" and is based on the hypothesis that variation should be higher among low fitness individuals. This section was removed.

l. 229-31 the argument about F_{st} may indeed be convincing but please note that in success-selected lines selection is expected to act in the same direction on deleterious variants, which would prevent differentiation via drift in these variants, thus slowing down the increase of overall F_{st} (this effect may be small as these are mostly low frequency polymorphisms, but still). In the failure-selected lines on the contrary, frequencies of both alleles in these loci are more free to vary via drift.

l. 237 the claim that selection depletes variation without a reduced N_e is too bold in my opinion

l. 249-51 these are rather weakly supported speculations

These comments are all focused on the F_{ST} values and our suggestion that N_e was not different between selection regimens. We have removed this suggestion and been more circumspect. Indeed, we accept that the F_{ST} values indicate the N_e of the success-selected lines were lower than the failure-selected lines, and have identified this as a key area for future research.

I. 247-8 wouldn't that be against the predictions of genic capture hypothesis?

This comment was removed with the last section.

compare I. 84 and 308: four vs one generation of relaxed selection. Does this mean that selection/relaxed selection episodes were intermixed during 14 generations of selection? Please provide more explanation.

Yes, we have amended this for clarity. There were three generations of relaxed selection intermixed with the 14 generations of selection. (I277-278)

I. 315 -316 libraries were constructed and sequenced – what was library the type?

We have expanded our sequencing section and added the library type (Illumina gDNA shotgun library preparation).

I. 325 loci -> reads

Fixed.

I. 367-74 this reasoning may have allowed you to estimate the ration, but it'd be much cleaner if you had estimates of actual N_e in your lines

The F_{ST} values from the simulations suggest N_e in the range of 33-42, but we do not have actual N_e values.

I. 430 I don't understand why you think this was a conservative approach – you probably simulated weaker overall drift than in the experiment so I wouldn't call this approach conservative.

We have removed this comment.

Reviewer #3 (Remarks to the Author):

I would like to thank the authors for the revised edition of the manuscript. The majority of my initial comments have been addressed, but I have a few additional suggestions/questions.

Line 124: 'density of SDVs and DiffStat loci'. I'm assuming this is referring to the density along the chromosome arms from Figure 1? If so, please state this or move this statement so that it immediately follows the sentence where you refer to Figure 1.

We have moved this sentence as suggested (I121-124).

Line 183: Since no other section of the supplemental methods is identified I suggest removing the (SM7) from the sentence.

Removed.

While I understand that no GO terms showed significant enrichment, it would be nice to have an idea what the 763 GO terms were associated with. Is it possible to put this information in a supplemental table? Also, in my experience 57 loci is a relatively small data set for enrichment analysis and I'm wondering if this is why you didn't observe enriched pathways.

We have amended this section: "While this result aligns with the genic capture hypothesis in suggesting that a broad array of molecular pathways (condition) were associated with divergence, it is also possible that the lack of significant GO terms is a simple consequence of having relatively few SDVs to test." Our data file (uploaded with the manuscript, and to be deposited on datadryad.org) contains gene names for FlyBase so that the interested reader can look up the functions of genes associated with divergence.

It seems as though the majority of the discussion focuses on the new analyses associated with examining how variation in N_e might be impacting the results. However, all of these results have been placed in a supplemental file. If this is now the main point of the manuscript, these data need to be moved to the main body of the paper.

I also find that the discussion is rather narrow in its focus, particularly regarding how N_e might impact the results. I would suggest the authors try to broaden this section (and perhaps the introduction) and discuss how these data support the genetic capture hypothesis in contrast to alternative hypotheses regarding the maintenance of genetic variation under sexual selection.

We agree with the reviewer that the focus shifted too much to the N_e simulations. We have (slightly) expanded our introduction to discuss, in more detail, alternative explanations for the maintenance of genetic variation and the genetic architecture of what we might expect to see under alternative models. We have also included a discussion paragraph that focuses more on the maintenance of genetic variation. (I207-218)

Line 318: should be '100 bp paired-end sequences were constructed and sequenced using an Illumina HiSeq2500 platform'.

Fixed as advised. (I297)

Line 325: What version of bwa?

Fixed. Version 0.7.12-r1039. (I301)

Lines 321: "total number of paired-end reads ranged from 21.5 – 23.2 million." From the sup table it seems as though you are averaging around 43 million per line. If the reads in the table are the total per sample (including forward & reverse), please state that to prevent confusion.

We have clarified this in the Table caption.

Lines 369: "The expected number of DiffState loci....."

I think that this is an important finding in regard to how the data set are likely associated with selection rather than just drift and should be included in the main results section.

We have performed these simulations again with $N_e=50$ and $N_e=33$. Our observed value now falls just within the 95% confidence intervals for one set of simulations.

Reviewer #2:

Remarks to the Author:

The revised version has addressed all the remaining issues and I'm happy to recommend acceptance. There is one minor issue remaining: in l. 304 please say what was the library type (TruSeq, Nextera, etc.) was used, and the sentence in l. 306 need to be corrected (something like: "100-bp paired end sequences were obtained using an Illumina HiSeq2500 platform").